# DGCB-Net: Dynamic Graph Convolutional Broad Network for 3D Object Recognition in Point Cloud

**Yifei Tian [1,2], Long Chen [2], Wei Song [1,*], Yunsick Sung [3] and Sangchul Woo [3]**

1 School of Information Science and Technology, North China University of Technology, Beijing 100144, China; yb87403@um.edu.mo
2 Department of Computer and Information Science, University of Macau, Macau 999078, China; longchen@um.edu.mo
3 Department of Multimedia Engineering, Dongguk University-Seoul, Seoul 04620, Korea; sung@dongguk.edu (Y.S.); woo.si@dgu.ac.kr (S.W.)
* Correspondence: sw@ncut.edu.cn; Tel.: +86-135-5228-4980

**Abstract:** 3D (3-Dimensional) object recognition is a hot research topic that benefits environment perception, disease diagnosis, and the mobile robot industry. Point clouds collected by range sensors are a popular data structure to represent a 3D object model. This paper proposed a 3D object recognition method named Dynamic Graph Convolutional Broad Network (DGCB-Net) to realize feature extraction and 3D object recognition from the point cloud. DGCB-Net adopts edge convolutional layers constructed by weight-shared multiple-layer perceptrons (MLPs) to extract local features from the point cloud graph structure automatically. Features obtained from all edge convolutional layers are concatenated together to form a feature aggregation. Unlike stacking many layers in-depth, our DGCB-Net employs a broad architecture to extend point cloud feature aggregation flatly. The broad architecture is structured utilizing a flat combining architecture with multiple feature layers and enhancement layers. Both feature layers and enhancement layers concatenate together to further enrich the features' information of the point cloud. All features work on the object recognition results thus that our DGCB-Net show better recognition performance than other 3D object recognition algorithms on ModelNet10/40 and our scanning point cloud dataset.

**Keywords:** point cloud analysis; 3D object recognition; broad learning system; dynamic graph convolution

## 1. Introduction

3D object recognition from clustered scenes is a popular research topic benefit for massive computer vision applications, such as intelligent surveillance, mobile robots, target tracking, and remote sensing [1]. Object recognition results provide supplementary information for these applications to perceive and understand the environmental situation. 3D object models captured by Light Detection and Ranging (LiDAR) or other range sensors consist of massive point clouds with rich geometric and shape information. Point cloud technologies based on LiDAR devices are widely utilized in many application platforms, for example, airborne laser scanning (ALS), mobile laser scanning (MLS), and terrestrial laser scanning (TLS). 3D object recognition based on LiDAR point clouds collected from these platforms is a meaningful research topic [2].

Although the significant advantage of 3D point clouds is its numerous original geometric and topology information, the disadvantage is that its inherent characteristics always cause a series of bottleneck problems in feature extraction [3]. The inherent characteristics mainly include uncertain vertex topology, uneven point density, arbitrary point count, and unfixed point permutation [4]. To avoid the influence caused by these characteristics, point cloud pre-processes, or registration steps are required in the majority of 3D object recognition methods [5].

For sampling uniform features from 3D object models, some researchers use volumetric models with a pre-defined resolution to resample and initial the point cloud [6]. Multiview transform is another common method to transform the uncertain geometric structure into fixed temporal or frequency domains [7]. These projection-based feature representation models contain a certain degree of information loss during transforming processes. To avoid information loss existing in these kinds of projection-based models, point-based feature representation methods have gradually been adopted by most researchers in recent years. Through utilizing weight-shared multiple-layer perceptrons (MLPs), pointwise features are extracted from the point cloud directly without any preprocessing steps [8]. Only global features that fused through using symmetrical functions (e.g., max pooling) are not sufficient for object recognition applications that require high recognition performance. Thus, some local feature grouping methods (e.g., farthest point sampling, *k*-nearest neighboring) are adopted to extract local point cloud features.

The graph structure is a popular representation that can efficiently extract geometry and topology information from the point cloud. Inspired by the dynamic graph convolutional neural networks (DGCNNs) in the deep learning domain [9], this paper developed a dynamic graph convolutional broad network (DGCB-Net) that maps and flatly extends graph features to enrich point cloud features. After aggregating dynamic graph features obtained from convolutional networks, we utilized a broad structure to rich graph information, thus that the recognition performance was better than those only using graph convolutional features. One significant difference between our DGCB-Net and classic CNNs was the proposed broad structure structured by multiple additional feature and enhancement layers, which maintained an independent relationship with each other. Because the convolutional results in the previous layer were the inputting data of the next layer, the relationship among contiguous layers in CNNs was close and interdependent. To provide additional information from different perspectives, adopting independent feature and enhancement layers was a suitable solution to enhance point cloud features.

To make up for the intimate relationship among the aggregation features extracted from the CNNs, the broad structure was adopted to detect and perceive more potential information from different perspectives. The broad structure in our system consists of multiple feature layers and an enhancement layer that extended flatly with an independent mapping relationship. During the feature layer mapping process, the weights and bias were generated randomly and optimized independently through dozens of iterations. The optimizing process of the feature layers follows the principles of sparse autoencoders. The enhancement layer is also a very important part of our broad structure that further expands the broad network of our proposed DGCB-Net. The parameters in enhancement layers are composed of a series of orthogonal basis, which enhances the capability of feature extraction in a high hierarchy. The training process of the output layer is crucial, where our model adopted the ridge regression method to obtain the optimal solution without any time-consuming process in the gradual parameter adjustment. Thus, the training process of the broad structure was much simpler than that of the convolutional layers. Distinctly, the broad structure of the DGCB-Net model was intuitive, concise, and owning the efficient capabilities of automatic feature extraction and object classification.

The primary contributions of this paper are listed as follows:

1. A DGCB-Net architecture adopts a broad method to improve the recognition performance of deep learning structures. This way, the model capabilities of both feature extraction and object recognition are strengthened.
2. The object recognition performance of the proposed DGCB-Net consists of improvement in both open point cloud dataset ModelNet10/40 and our collected outdoor common objects. When the inputting point counts are uniformly downsampled, the recognition results are especially better than the other popular methods, which means our proposed DGCB-Net shows robust performance for sparse point clouds.
3. Pioneeringly, we bring the broad structure into the point cloud processing domain to enhance the convolutional features of point clouds. Besides, the proposed broad

structure is lightweight with fast training speed, which means it only requires a few additional time and calculation consumptions to produce an efficient improvement for the deep learning model.

The paper is organized as follows. Section 2 investigates related work about LiDAR point cloud processing methods and popular object recognition models. Section 3 describes the structure and theory of our proposed DGCB-Net model. Section 4 illustrates our experimental platform, LiDAR point cloud library, and object recognition results estimated on ModelNet10/40 and outdoor object samples collected by a Velodyne-32E LiDAR. Section 5 is the conclusion of our paper.

## 2. Related Works

This section described several typical inherent characteristics of point clouds and common bottleneck problems that existed in object recognition. Some classic and useful hand-crafted descriptors were briefly analyzed about their advantages and disadvantages. Besides, several innovative and popular state-of-the-art object recognition methods inspired by deep learning algorithms were discussed and summarized.

Point clouds generated by CAD models or collected by range sensors possessed some inherent characteristics, like uncertain topology, density, point count, permutation, and noise. Because the efficiency of feature extraction seriously affected the object recognition accuracy, these inherent characteristics were widely discussed and researched in the point cloud domain. For example, the uncertain vertex topology characteristic resulted in complex neighborhood area division and time-consuming neighboring points research [10]. Point clouds with uncertain density always required a density balancing step as a preprocess before consequent feature extraction and object recognition [11]. Besides, different 3D object models did not contain the same point count. In this way, point clouds were not available to be analyzed directly by unified kernels or descriptors. In addition, the relationship between point order and their spatial distribution was not indefinite. The unfixed point permutation characteristic caused large computational consumption in neighboring point research and local feature extraction [12]. This way, to avoid the disadvantages caused by these inherent characteristics, copious classic descriptors were proposed and widely analyzed by researchers.

Classic descriptors mainly focused on key point search, local feature description, surface shape retrieval, and category recognition [13]. Yang et al. [14] proposed a triple orthogonal local depth images (TOLDI) descriptor to perceive point clouds' local features based on a local reference frame (LRF). If the TOLDI descriptor is used, a series of local geometry information is required to be calculated, such as normal and projection vector in original 3D space, and neighboring point projection and local depth distance after virtual view transformation in the LRF. Similar to the local voxelized structure (LoVS) descriptor proposed by Quan et al. [15], LRF was also constructed as the first step to transform neighbor points from global to a local coordinate system with rotation invariance. Through projecting neighbor points into a binary cubic volume, local features were represented as a zero/one sequence with implicit meaning. These classic local feature descriptors always required LRF construction and neighbor points' projection, thus that the speed performance was highly relative with the neighboring area definition [16].

Inspired by the remarkable achievement of image processing in the deep learning domain, some researchers combined classic descriptors and deep learning algorithms to increase feature extracting efficiency [17]. Without hand-draft feature selection, Han et al. [18] proposed an unsupervised method to extract point cloud features through utilizing stacked sparse auto-encoder (SSAE). After mapping object point clouds into a pre-defined 3D voxel model, they adopted four fixed views to sample point distribution features. Automatically, both local and global features were extracted based on a voxel model after an unsupervised training process. Theoretically, voxel models with higher resolution have stronger information expressing capability [19]. Limited by the computing performance of the experiment platform, the most suitable voxel model resolution of

different 3D objects is barely set as an extreme highest value. Corresponding to different scenes and applications, the most suitable voxel resolution is relative to individual object size and point density [20].

Traditional convolution operations were always applied on a regular grid structure with a concise neighbor grid relationship (e.g., neatly arranged pixels in images) [21]. However, vertexes in point clouds were arranged arbitrarily, thus, that point clouds are not available to be processed directly by classic CNNs. To overcome the arbitrary topology relationships among neighbor points, Groh et al. [22] proposed a flex convolution method to map arbitrary Euclidean distance from a continuous to a discrete domain using a grid shape model and a discrete function. This way, local point cloud features were converted as a series of regularly arranged values thus that convolution operation was available to work on point clouds based on the voxelization strategy. However, the resolution selection problem still troubled these convolutional-based point cloud recognition algorithms [23].

If these voxel models or convolutional operators utilized a single resolution, the feature information obtained from point clouds was insufficient. Liu et al. [24] proposed DensePoint that used a series of multi-scale operators in different convolutional layers to obtain rich local geometry features. Based on the feature maps generated by different sizes of convolutional operators, they concatenated feature maps together by using a long-range connection method. Compared with basic convolutional models, DensePoint showed better overall accuracy on the open dataset ModelNet 40. Different from simple convolution operation working on regular grids, Esteves et al. [25] proposed a spherical CNN architecture that represents object point clouds by a spherical model. Based on spherical convolution and pooling operations, spherical CNN exploited distance and surface normal without any pre-defined resolution. Besides, another obvious advantage of the spherical CNN was the insensitive characteristic for point cloud orientation and rotation angles in three axes. Although these convolution-based algorithms maintain better-recognizing performance than voxel-based algorithms, convolution-based models also required pre-defined continuous or discrete kernels to process unstructured point clouds in 3D spaces. Whether continuous or discrete convolutional kernels, the design was more complex than that of 2D data (e.g., image).

Graph-based point cloud processing algorithms were widely researched in recent years because of their strong capability in topology feature extraction [26]. In graph-based object recognizing models, each point of the point cloud is considered as a vertex in a graph structure, thus that the topology information was represented better than other models. Shen et al. [27] proposed a kernel correlation network (KCNet) to realize semantic segmentation and object classification based on graph architecture. In this method, a *k* nearest neighbor (k-NN) method was adopted to define neighborhood area for searching neighboring points. Through utilizing a graph max pooling and a kernel correlation, the vertex topology features were extracted from neighboring points. Although graph-based algorithms did not have to consider the convolution resolution, the neighboring point search was still a time-consuming procedure in the graph convolutional process [28].

Because point-based models were available to deal with point clouds directly in 3D space, point-based feature representation became popular in object recognition domains [29]. Hua et al. [30] developed a pointwise CNN framework that employed convolution operation on each point to obtain pointwise features. Using several convolutional layers to expand the point cloud from 3 to 36 dimensions, fully connected layers were added in the end to realize object classification. The simple convolution and fully connected layers alone were inadequate, thus that the performances of both semantic segmentation and classification were not satisfied. Qi et al. [8] proposed PointNet architecture with a transform model and weight-shared MLPs to extract pointwise point cloud features. Similar to Hua's model, PointNet also employed several full connection layers directly on the last mapping layer to realize the classification function. To supplement point cloud features, the feature map obtained by the last layer and DensePoint also concatenated features from different convolutional layers together to form the following recognition

foundation. Through using this kind of feature concatenation approach, object recognizing performance realized a certain degree of improvement.

Different from deep learning algorithms, Chen et al. [31] adopted a broad learning system with significant advantages of the flexible structure and fast training speed. Based on the broad architecture, Zhang et al. [32] added a graph convolution structure in it to realize good performance in emotion recognition. Inspired by the graph and broad fusion architecture, this paper proposed DGCB-Net to improve object types from point clouds within affordable computation consumption.

## 3. Object Recognition Method from 3D Point Clouds

As discussed in Section 2, raw point clouds sampled from CAD models or collected by range sensors own special characteristics, especially their permutation and uncertain structures. To overcome the above difficulties, our proposed DGCB-Net employed a view transform network and several edge convolutional layers. In each edge convolutional layer, we applied the *k*NN method to generate subgraphs with abundant local features and used weight-shared multiple-layer perceptrons to extract them. This section introduces the whole architecture of our proposed DGCB-Net model, contains the detailed process of graph generation, and definitions of edge convolutional layers. Our system was roughly divided into 2 parts, the feature extraction part to aggregate features and the broad learning network part to recognize objects. Their details are described in the following Sections 3.1 and 3.2.

### 3.1. Graph Feature Generalization and Aggregation

In our system, *m* edge convolutional layers were stacked together to extract point information from the data $\rho^l \in \mathbb{R}^{n \times d_l}$. The variable *l* is the edge convolutional layer index, $l \in [1, L]$. Matrix $\rho^{l-1}$ means the input data of the *l*th convolutional layer, for example, the $\rho^0$ is the original point cloud as the input of the 1st convolutional layer. The input feature of $\rho^l$ contains *n* elements $p_i^l$ as $p_i^l \in \rho^l$, $i \in [1, n]$. The variable *n* is the point count, which is uncertain among different object samples. The variable $d_l$ is the dimension of the vector $p_i^l$ as $p_i^l \in \mathbb{R}^{d_l}$.

In the *l*th layer, the mapping relationship between its input data $\rho^{l-1}$ and its the output $\rho^l$ is defined as $\Phi^l : \mathbb{R}^{d_{l-1}} \to \mathbb{R}^{d_l}$. Each layer executes a *k*NN algorithm first to generate a dynamical graph structure $G^l = \left(\rho^l, E^l\right)$ for concatenating edge topology information, where the vertex $\rho^l$ and the edge $E^l \subseteq \rho^l \times \rho^l$. As shown in Figure 1a, the local feature of the center point $p_i^l$ is related with its *k* nearest neighboring points $p_j^l$, $j \in [1, k]$. The edge feature $e_{i,j}^l$ between point $p_i^l$ and $p_j^l$ is computed according to Equation (1) with the edge mapping function $\psi^l$.

$$e_{i,j}^l = \psi^l \left( p_i^l, p_i^l - p_j^l \right) \tag{1}$$

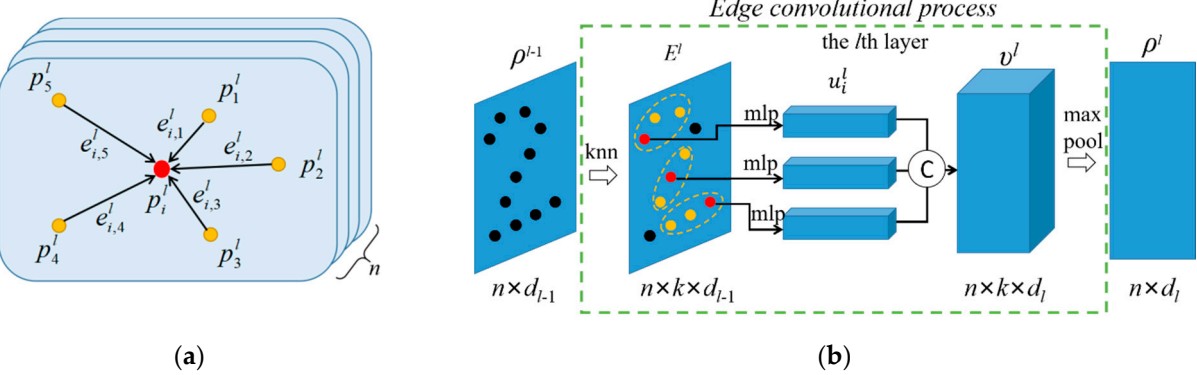

(a)  (b)

**Figure 1.** Point and edge feature transformation in the *l*th graph convolutional layer. (**a**) Neighboring points with edges, (**b**) feature transformation in the *l*th convolutional layer.

Figure 1b illustrates the total process of the *l*th edge convolution layer, where we adopted a series of weight-shared MLPs to exact the edge features. After executing the *k*NN method, the input data $\rho^{l-1}$ generates the edge feature tensor $E^l$ as $E^l \in \mathbb{R}^{n \times k \times d_{l-1}}$. Equation (2) describes the generation of feature $u_i^l$ from edge feature $e_{i,j}^l$ based on the edge convolutional operator $\phi(.)$. Similar to the classic convolutional principle in the deep learning domain, the edge convolutional operator $\phi(.)$ the *l*th convolutional layer consists of weight $w_i^l \in W^l$ and bias $b_i^l \in b^l$ to transform features as $\mathbb{R}^{d_{l-1}} \times \mathbb{R}^{d_{l-1} \times a_l} \to \mathbb{R}^{a_l}$, $W \in \mathbb{R}^{d_{l-1}} \times \mathbb{R}^{d_l}$. The implementation details of most MLP in the point cloud domain are realized by convolutional operators.

$$u_i^l = \underset{j \in [1,k]}{\phi^l} \left( e_{i,j}^l w_i^l + b_i^l \right) \tag{2}$$

All the features $u_i^l$ are concatenated as the feature tensor $v^l$, defined as $v^l \in \mathbb{R}^{n \times k \times d_l}$. Considering that the model requires permutation-symmetric compression and aggregation, a max aggregation operation $\varphi^l$ is employed on the feature tensor $v^l$ to obtain the feature $\rho^l$ in the *l*th edge convolutional layer in Equation (3).

$$\rho^l = \varphi^l \left( \underset{j \in [1,k]}{\phi^l} \left( E^l W^l + b^l \right) \right) \tag{3}$$

Figure 2 illustrates the implementation details of feature generation and aggregation by using a view transform network and weight shard MLPs achieved by convolutional layers, batch normalization layers, leaky ReLu activation functions, and max pool layers. After executing *L* layers of edge convolutional operations, the point features of the object sample were transformed from dimension $n \times d_0$ to $2d_L$. Through using a back-propagation method, these edge convolutional layers were pre-trained with feature extraction capability. The cost function was computed according to the following Equation (4). The $Y$ and $\overline{Y}$ are the ground truth and the predicted output of object labels, which contains $N$ samples with $M$ classes, respectively.

$$J(Y, \overline{Y}) = -\frac{1}{N} \sum_{s=1}^{N} \sum_{c=1}^{M} y_{s,c} \log\left(\overline{y}_{s,c}\right) \tag{4}$$

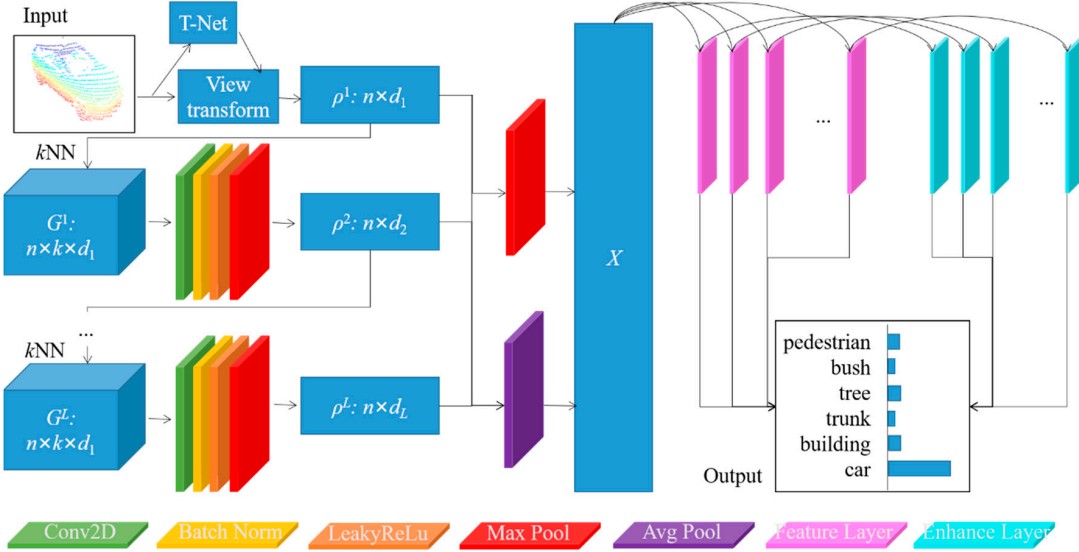

**Figure 2.** The framework of the proposed Dynamic Graph Convolutional Broad Network (DGCB)-Net with feature extraction and object recognition modules.

When the pre-training process was finished, our proposed DGCB-Net model concatenated the intermediate features obtained from different convolutional layers to increase the feature utilization efficiency. We employed both max and average pooling operators to compress point features $\rho^l$ ($l \in [1, L]$) into as $X$, which was considered as the input data of the following broad network.

### 3.2. Broad Network Construction

Our proposed broad network structure was inspired by the basic broad learning system [31], as shown in Figure 3. Different from popular deep learning models, the broad learning system expanded the model structure flatly instead of stacking them layer-by-layer. The most significant advantages of the broad learning system were the fast training speed and not inferior fitting performance. Inspired by the broad learning system, we pioneered the broad structure into the deep learning models as a feature enhancement component to improve the recognition performance of convolutional neural networks.

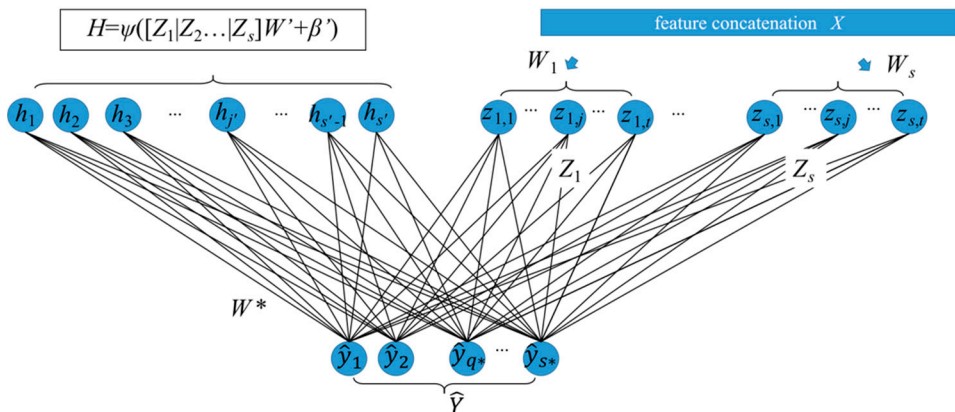

**Figure 3.** The broad network with concatenating input.

As shown in Figure 3, all object features obtained by these edge convolutional layers were concatenated together as $X$, $x_i \in X$, $i \in [1, d^L]$, where variable $d^L$ is computed as $d^L = 2 \times \sum\limits_{l=1}^{L} d_l$. For enhancing the point cloud feature, $X$ was mapped into $m$ feature layers $Z_j$ with individual projection relationships among each layer. Different from the deep learning models that employed the gradient descent method on the training process, the parameter optimizing and fine-tuning processes of the broad structure were faster and more concise. In the feature layer, a series of mapping matrix $W_j$ were randomly initialized to project $X$ to $m$ feature layers $Z_j$ according to Equation (5). Different from the relationships among convolutional layers, feature layer $Z_j$ was not influenced by the last layer $Z_{j-1}$ in broad networks. In other words, the input data of all the feature layers $Z_j$ were directly connected with the input data $X$, which means the projecting operation between features $X$ and different feature layers $Z_j$ are independent of each other.

$$Z_j = \xi(XW_j + b_j) \tag{5}$$

Our broad structure adopted a sparse autoencoder to fine-tune their weight matrix $W_j$ instead of the popular gradient descent method. The fine-tuning process was transformed into a convex optimization problem, as shown in Equation (6), which target was to obtain the sparse and efficient weight matrix $W_j$ in feature layers. The matrix $W_j$ in Equation (6) contained an additional column vector with value 1 as an augmented matrix to multiple

with the bias item. Through defining the optimization problem as the *lasso* format, the near-optimal solution $W_j$ was available to be solved by many methods.

$$W_j = \underset{W_j}{\mathrm{argmin}} \frac{1}{2} \|ZW_j - X\|_2^2 + \alpha \|W_j\|_1 \tag{6}$$

Through using the L1 regularization as a penalty term during the optimization process, the obtained matrix $W_j$ was composed of meaningful values and 0 values. Lots of algorithms, such as the alternating direction method of multipliers (ADMM), orthogonal matching pursuit, and fast iterative shrinkage-thresholding algorithm, were widely discussed and utilized for solving a convex optimization problem. We used the ADMM algorithm as an example to describe how the matrix $W_j$ was computed by several times of iteration. According to the ADMM algorithm, the total objective function was equally divided into 2 parts $f_1(W_j)$ and $f_2(W_j)$ as shown in Equation (7).

$$\min f(W_j) = \min f_1(W_j) + f_2(W_j) = \min \frac{1}{2} \|ZW_j - X\|_2^2 + \alpha \|W_j\|_1 \tag{7}$$

Next, the optimization problem was transformed into a global variable consensus optimization problem with the equality constraints written as Equation (8). The constraint condition requires that the local variable $W_j$ in the objective function is consistent with the global variable $\delta$.

$$\min f_1(W_j) + f_2(\delta), \; s.t. \; W_j - \delta = 0. \tag{8}$$

The ADMM algorithm implement by dozens of the iteration process, where the variable $W_j$ and $\delta$ are alternately updated as shown in Equation (9). Because the objection function of the optimization problem can be decomposed as $f_1(W_j)$ and $f_2(W_j)$, alternate updating the two variables $W_j$ and $\delta$ was available to be adopted as the solution. The variable $u$ is the dual variable, $\rho > 0$ is the penalty coefficient.

$$\begin{cases} W_j^{k+1} = \underset{W_j}{\arg\min}\left( f_1(W_j) + (\rho/2)\|W_j - \delta + u^k\|_2^2 \right) \\ \delta^{k+1} = \underset{\delta}{\arg\min}\left( f_2(\delta) + (\rho/2)\|\delta - W_j^{k+1} - u^k\|_2^2 \right) \\ \qquad u^{k+1} = u^k + W_j^{k+1} - \delta^{k+1} \end{cases} \tag{9}$$

Through dozens of updating iterations, we obtained the mapping matrix $W_j$ with sparse characteristics. The $s$ groups of feature layers $Z_j$ were concatenated flatly as $Z^s \in \mathbb{R}^{N \times st}$ as the input data of the enhancement layer $H$. The enhancement layer $H$ used a series of orthogonal basis $W'$ to improve the expressiveness of features. The implementation process of the enhancement layer is easy, and the generation speed was faster. Then, the enhancement layers $H$ were computed from $Z^s$ by a mapping matrix $W'$ and bias vector $\boldsymbol{\beta}'$ according to the Equation (10). The weight matrix $W'$ and bias $\beta'$ in the enhancement layers obeyed the orthogonal relationship. A nonlinear function $\zeta$ was used to project the generated features into a relatively uniform interval. The enhancement layer $H$ contained $s'$ enhancement nodes $h_{j'}$ denoted as $h_{j'} \in H, j \in [1, s']$.

$$H = \zeta\left(([Z_1|Z_2|\ldots|Z_s])W' + \beta'\right) \tag{10}$$

The reason for setting the enhancement layer was to improve the model's stability and feature extraction capability. Then, both feature $Z^s$ and enhancement layers $H$ were extended together as feature matrix $A$ in a flat way. As shown in Equation (11), the output $\hat{Y}$ of our broad network was obtained by multiplying feature matrix $A$ and a mapping matrix $W^*$. The dimensions of feature matrix $A$ and weight matrix $W^*$ were $\mathbb{R}^{N \times (st+s')}$

and $\mathbb{R}^{(st+s')\times s*}$, respectively. The detailed feature generation and model structure of our proposed DGCB-Net is illustrated in Figure 3.

$$Y = [Z^s|H]W^* = AW^* \tag{11}$$

For calculating the target mapping matrix $W^*$, this paper employed a ridge regression algorithm to obtain an optimal solution. Through using the biased estimation regression method, a reliable and realistic optimal result was obtained at the expense of partial information and reduced accuracy. As shown in Equation (12), our model's optimal target was narrowing the L2 distance between the predicted result $AW^*$ and ground truth $Y$, which means the object labels in our application. Meanwhile, an L2 regularization term $\lambda\|W^*\|_2^2$ was added in this objective function to control the weights $w^*$ that belonged to the mapping matrix $\boldsymbol{W^*}$ located in a relatively small value. This way, the obtained optimal solution can alleviate the over-fitting problem to a certain extent.

$$\underset{W^*}{\mathrm{argmin}} \; : \; \|AW^* - Y\|_2^2 + \lambda\|W^*\|_2^2 \tag{12}$$

In the process of solving regression coefficients, it was necessary to consider whether the characteristic matrix was invertible. When the number of features $d^L$ is greater than the number of samples $N$ (matrix $X$ is not full-rank), standard linear regression cannot solve the optimal solution. In other words, the inverse of the matrix $A^TA$ shown in Equation (13) cannot be solved.

$$W^* = A^+Y = \left(A^TA\right)^{-1}A^TY \tag{13}$$

Thus, broad learning networks use ridge regression to deal with the problem of features' numbers bigger than the samples' numbers. By adding a disturbance term $k\boldsymbol{I}$, as shown in Equation (14), the matrix $X^TX+kI$ becomes non-singular, thus that its pseudo-inverse can be computed. The variable $k$ is a super parameter that influences the identity matrix $\boldsymbol{I}$, whose diagonal consists of a series of the number 1.

$$W^* = \left(A^TA + kI\right)^{-1}A^TY \tag{14}$$

This way, through using the computed weight matrix $W^*$, our proposed DGCB-Net model can realize the automatic feature extraction and object classification. The broad structure initiative combines with deep learning to achieve a certain degree of feature enhancement. The broad structure was simple to implement, and good performance can be achieved with only a little additional time on the deep learning training process. On account of the broad architecture, our DGCB-Net model computed the weight and bias parameters without any gradient descent computation in the training process. If the trained model was not satisfied with the required performance, our proposed DGCB-Net adopted an incremental learning method for remodeling the broad structure. The trained part did not need a retraining process, new feature and enhancement layers were inserted flatly to expand the existing structure.

## 4. Experiments and Analysis

This section estimates our DGCB-Net on several public datasets ModelNet10 and a dataset that was collected by our Unmanned Ground Vehicle (UGV) platform. Figure 4a is our data collection platform that is an EU260 unmanned vehicle produced by the BAIC Motor Corporation and carried with an HDL-32E Velodyne LiDAR sensor. Our experiment ran on a computer carrying Intel® Xeon CPU E5-1650 CPU with 64.0 GB RAM and a GeForce GTX 1070 GPU with 8.0 GB memory. We employed the open deep learning architecture Pytorch 1.5 and a GPU development toolkit CUDA 10.2 to train the pre-training model.

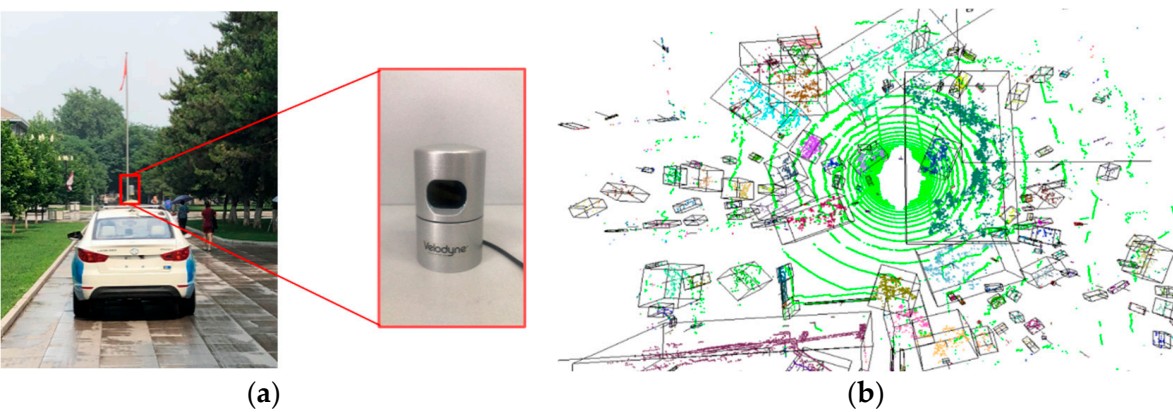

**Figure 4.** Our dataset collection platform and dataset generation from outdoor scenes, (**a**) UGV with LiDAR sensor, (**b**) outdoor object that segment from the scene.

Figure 4b is a point cloud of a whole outdoor scene captured by the LiDAR sensor, which contains multiple separated obstacles that generate our collected datasets. Table 1 shows the statistical data of datasets that were used in our experiment, including the numbers of training samples, testing samples, average point numbers, and classes. Dataset ModelNet10 contained 10 classes, containing 3991 training samples and 908 testing samples. Dataset ModelNet40 contained 40 classes with 9840 training samples and 2468 testing samples. All the samples in both ModelNet10 and ModelNet40 owned 1024 points, which were uniformly sampled from standard CAD models. The outdoor object dataset consisted of LiDAR point clouds collected from the real scene. Different from the point cloud generated from standard CAD models, LiDAR point clouds were the surface of outdoor obstacles without inside or back parts. Thus, to estimate the recognition performance on real LiDAR point clouds, we collected 1010 real samples for model training and 463 for testing. The average point number of our collected dataset that was named as an outdoor object was around 415, which was slightly less than the 490 of the Sydney datasets.

**Table 1.** Dataset Description.

|  | **Modelnet10** | **Modelnet40** | **Outdoor Object** |
|---|---|---|---|
| Training Samples | 3991 | 9840 | 1010 |
| Testing Samples | 908 | 2468 | 463 |
| Avg. Point Number | 1024 | 1024 | 415 |
| Classes | 10 | 40 | 6 |

### 4.1. Modelnet10 and Modelnet40

**Algorithm Comparison** Modelnet10 and ModelNet40 were 2 popular point cloud datasets to estimate object recognition algorithm. Various object recognition methods were developed and tested based on the 2 datasets, which were ideal datasets generated by CAD models. As shown in Table 2, the recognition performance of our proposed DGCB-Net algorithm was slightly higher than other classic algorithms. For example, it was conveniently observed that the performance of our algorithm was better than point-based input according to the accuracies of the PointNet [8], PointNet++ [33], and Pointwise-CNN [30]. That was because point-based algorithms (e.g., PointNet and Pointwise-CNN) only consider global point features without local geometry information, which means the input of these kinds of models were just raw point coordinates. Their convolutional-like operators applied on global point coordinates, and global aggregation operation (e.g., max/average pooling) was adopted to generate global features. This way, no local features were the most significant bottleneck problem that affected the recognition performance of these pointwise algorithms that only contained global features. The accuracy rates of ModelNet40 obtained from Pointwise-CNN and PointNet algorithms were less than 90%.

To supplement the local features, PointNet++ adopted a single-scale grouping that divides point clouds into different small subspaces with the local feature aggregation function, thus that the accuracy upgraded to 90.7%.

**Table 2.** Accuracy of ModelNet10/40 under popular recognition algorithms.

| Method | | Input | Modelnet40 (Accuracy %) | Modelnet10 (Accuracy %) |
|---|---|---|---|---|
| Pointwise-based Networks | PointNet [8] | point | 89.2 | - |
| | PointNet++ [33] | point | 90.7 | - |
| | Pointwise-CNN [30] | point | 86.1 | - |
| Voxel-based Networks | VoxNet [34] | voxel | 83 | 92 |
| | 3DShapeNets [35] | voxel | 77.3 | 83.5 |
| | BV-CNNs [36] | voxel | 85.4 | 92.3 |
| | ORION [37] | voxel | - | 93.8 |
| Image-based Networks | MVCNN [38] | image | 90.1 | - |
| | DeepPano [39] | image | 82.5 | 88.7 |
| Graph-based Networks | ECC [26] | graph | 87.4 | 90.8 |
| | DGCNN [9] | graph | 92.2 | - |
| | DGCB-Net (Our) | graph | 92.9 | 94.6 |

Voxel-based and image-based algorithms own similar point cloud processing flow, where unstructured and uncertain ordered point clouds were transformed into a series of regular pixels or voxels in the initial phase. As shown in Table 2, the accuracies of the 2 voxel-based methods VoxNet [34] and 3DShapeNet [35] were around 77% and 84% on Modelnet40, which were around 10% less than graph-based algorithms. The BV-CNN [36] and ORION [37] models showed relatively better results than VoxNet [34] and 3DShapeNet [35] on ModelNet10. The average accuracies of image-based algorithms MVCNN [38] and DeepPano [39] were 90.1% and 82.5%, respectively. The main reason that caused pool performance in voxel-based and image-based algorithms was their coarse grain sampling resolutions of 2D/3D grids. A certain information loss exists in these information sampling processes, at the same time, the information loss is almost irreparable. This way, the rigid sample method causes some limitations in the recognition performance of voxel-based and image-based algorithms, which have become less researched in recent years.

Different from the poor performance in voxel-based and image-based algorithms, using graph structure to extract both local and global features significantly improved the recognition efficiency in the point cloud domain. Graph structure owns the better precepting capability than 2D/3D grids to extract topology information among center point and its neighboring points. Besides, graph-based algorithms do not exit information loss of point cloud sampling during graph structure generation. This way, graph-based algorithms own better recognition performance than voxel/image-based algorithms on both ModelNet10/40. We listed the 2 most popular graph-based algorithms ECC [26] and DGCNN [9], in Table 2. The ECC [26] algorithm uses a fixed graph structure with multiple resolutions before point cloud inputting. Whereas DGCNN generates dynamic graph structure in each convolutional layer, which means richer information is perceived from different graph spaces. Thus, the recognition performance of the graph-based network ECC [26] was slightly worse than the DGCNN [9] on both ModelNet10 and Modelnet40. Besides, the recognition accuracy of DGCNN was higher than other popular point-based, voxel-based, and image-based algorithms, which means the dynamic graph structure has stronger feature extraction capability than others.

Compared with the 2 graph-based algorithms that only adopted deep learning architectures, recognition results on ModelNet10/40 obtained by DGCB-Net were slightly improved. As shown in the last row of Table 2, our average accuracies on ModelNet10/40 were 94.3% and 92.8%, respectively. The main reason for our good performance was that we adopted a broad structure to enhance the recognition backbone flatly. Our algorithm combined graph features from all convolutional layers together and transformed them into multiple higher and more abstract feature spaces at the same time. These feature layers improved the discrimination of combining graph information from convolutional layers, thus that our proposed DGCB-Net algorithm owned the best recognition performance on the ModelNet10/40 at the same time.

To offer a detailed description of the recognition performance of our proposed DGCB-Net on the classic dataset ModelNet40, we gave accurate results of each class in Table 3. Compared with the DGCNN algorithm, our algorithm showed better performance in most object types. For example, our algorithm was effective for cars, cones, laptops, etc., where the F1 scores reached around 1.00. Besides, the recognition performance of table, plant, and TV stand obstacles were slightly better than that of DGCNN.

**Table 3.** Each class accuracy in ModelNet40 datasets.

| Network | Performance | Airplane | Bathtub | Bed | Bench | Bookshelf | Bottle | Bowl | Car | Chair | Cone |
|---------|-------------|----------|---------|------|-------|-----------|--------|------|------|-------|------|
| DGCNN | PR | 1.0 | 0.98 | 0.97 | 0.79 | 0.90 | 0.95 | 0.83 | 0.99 | 0.98 | 1.00 |
| | RC | 1.0 | 0.90 | 0.99 | 0.75 | 0.99 | 0.98 | 0.95 | 1.00 | 0.98 | 0.95 |
| | F1 | 1.0 | 0.94 | 0.98 | 0.77 | 0.94 | 0.97 | 0.88 | 0.99 | 0.98 | 0.97 |
| Ours | PR | 1.0 | 0.99 | 0.99 | 0.70 | 0.99 | 0.97 | 0.90 | 1.0 | 0.98 | 1.00 |
| | RC | 1.0 | 0.98 | 0.97 | 0.82 | 0.93 | 0.97 | 0.82 | 0.99 | 0.96 | 1.00 |
| | F1 | 1.0 | 0.96 | 0.98 | 0.76 | 0.96 | 0.97 | 0.86 | 1.00 | 0.97 | 1.00 |

| Network | Performance | Cup | Curtain | Desk | Door | Dresser | Flower Pot | Glass Box | Guitar | Keyboard | Lamp |
|---------|-------------|------|---------|------|------|---------|------------|-----------|--------|----------|------|
| DGCNN | PR | 0.61 | 0.95 | 0.79 | 0.95 | 0.80 | 0.20 | 0.97 | 0.99 | 0.95 | 1.00 |
| | RC | 0.70 | 0.95 | 0.88 | 0.95 | 0.86 | 0.30 | 0.95 | 1.00 | 0.95 | 0.90 |
| | F1 | 0.65 | 0.95 | 0.84 | 0.95 | 0.83 | 0.24 | 0.96 | 1.00 | 0.95 | 0.95 |
| Ours | PR | 0.70 | 0.90 | 0.90 | 0.85 | 0.92 | 0.10 | 0.96 | 1.00 | 0.95 | 0.85 |
| | RC | 0.67 | 0.82 | 0.84 | 0.94 | 0.72 | 0.18 | 0.97 | 0.98 | 0.95 | 1.00 |
| | F1 | 0.68 | 0.86 | 0.87 | 0.89 | 0.81 | 0.13 | 0.96 | 0.99 | 0.95 | 0.92 |

| Network | Performance | Laptop | Mantel | Monitor | Night Stand | Person | Piano | Plant | Radio | Range Hood | Sink |
|---------|-------------|--------|--------|---------|-------------|--------|-------|-------|-------|------------|------|
| DGCNN | PR | 0.95 | 0.99 | 0.97 | 0.81 | 1.00 | 1.00 | 0.88 | 0.80 | 0.98 | 0.94 |
| | RC | 1.00 | 0.98 | 1.00 | 0.81 | 0.95 | 0.95 | 0.80 | 0.80 | 0.97 | 0.85 |
| | F1 | 0.98 | 0.98 | 0.99 | 0.81 | 0.97 | 0.97 | 0.84 | 0.80 | 0.97 | 0.89 |
| Ours | PR | 1.0 | 0.95 | 1.0 | 0.74 | 0.95 | 0.94 | 0.88 | 0.75 | 0.97 | 0.95 |
| | RC | 1.0 | 0.97 | 0.95 | 0.91 | 1.0 | 1.00 | 0.87 | 0.94 | 1.00 | 1.00 |
| | F1 | 1.0 | 0.96 | 0.98 | 0.82 | 0.97 | 0.97 | 0.88 | 0.83 | 0.98 | 0.97 |

| Network | Performance | Sofa | Stairs | Stool | Table | Tent | Toilet | Tv Stand | Vase | Wardrobe | Xbox |
|---------|-------------|------|--------|-------|-------|------|--------|----------|------|----------|------|
| DGCNN | PR | 0.98 | 1.00 | 0.84 | 0.86 | 0.95 | 1.00 | 0.92 | 0.87 | 0.76 | 0.94 |
| | RC | 1.00 | 0.95 | 0.80 | 0.79 | 0.95 | 0.99 | 0.86 | 0.80 | 0.80 | 0.85 |
| | F1 | 0.99 | 0.97 | 0.82 | 0.82 | 0.95 | 0.99 | 0.89 | 0.83 | 0.78 | 0.89 |
| Ours | PR | 1.0 | 0.95 | 0.75 | 0.86 | 0.95 | 1.00 | 0.87 | 0.90 | 0.75 | 0.80 |
| | RC | 0.97 | 0.95 | 0.88 | 0.83 | 0.90 | 0.99 | 0.95 | 0.84 | 0.88 | 0.89 |
| | F1 | 0.99 | 0.95 | 0.81 | 0.85 | 0.93 | 1.00 | 0.91 | 0.87 | 0.81 | 0.84 |

**Parameter Detail** The structure of the DGCB-Net used for the ModelNet10/40 contained 4 feature generating layers, 4 2D dynamic graph convolutional layers, 1 2D combination convolutional layer, 33 feature layers, and 500 enhancement nodes. Based on the dataset, the *k* value was set as 20, and the edge features contained 6 dimensions. During each feature generating layer, the edge features and graph structures were recomputed, and the *k* nearest neighbors were ordered. The channels in the first 2 layers were 64, the 3rd layer owned 128 channels, the channels in the 4th layer were 256, and the last layer owned 1024 channels. After the first 4 convolutional layers, corresponding 2D batch normalization layers, LeakyReLU functions with 0.2 negative slopes, and max-pooling layers were following. The input of the 5th 1D convolution layer was the graph feature concatenation of the above 4 convolutional layers. After the first concatenation, both max and average 1D pool operators were adopted at the same time to downsample the graph feature and element the uncertain dimension of point counts. The concatenation of the 2 kinds of pooling results was the input of the feature layers in our broad structure. The results were projected into 33 feature layers under sparse autoencoders, which were separately initially obeying normal distribution and training under 50 iterations. The weights between 33 feature layers and 500 enhancement nodes were initialized as random normal distribution obeys an orthogonal constrain. The activation function in the feature layer was a linear function between [0, 1] and a tansing function in the enhancement layer. The features obtained from both the features layers and enhancement layer were concatenated together as the input of the output layer.

Based on the above feature extraction module, we estimated the recognition performance under different sizes of broad networks on the ModelNet10/40 (abbr. as MN10/40). Figure 5 contains 6 heat maps representing corresponding recognition accuracies, where high value was rendered in oxblood red, and low value was rendered in navy blue. The horizontal and vertical axes were feature node numbers in each feature layer and feature layer, respectively. The enhancement node numbers (N_E) were set as 100, 1000, and 2000 in the 3 columns. Most accuracy rates of ModelNet10/40 were located in the range from 90% to 95% under the given parameter options. In the first row, it was evident that an improved performance appeared in Figure 5a when enhancement nodes were set as 500. When increasing the enhancement to 1000, the main colors of the heat map were light orange and light blue. When the enhancement was increased to 2000, the overall performance was not good with a large area of blue. On the contrary, the MN40 dataset prefers high enhancement performance. As shown in the 2nd row, the dark orange mainly focused on Figure 5e,f. Besides, the inner trend of performance variance in each heat map was clear in the 2nd row, where the accuracy in the lower right corner was significantly higher than that of the upper left corner. This way, we can conclude that dataset MN40 relied on a wider architecture to obtain better performance, whereas MN10 favored a small broad network.

**Robust Estimation** We also estimated the robustness of our proposed DGCB-Net on point clouds with random input dropout. We selected a car sample from ModelNet40 as a template to visualize its object structure and point distribution under different point numbers 1024, 512, 256, and 128 in Figure 6a. When the points were dropped by 50%, at 512, the car still had a complete inside structure and clear contour information. When dropping 75% points in 256, there was a certain amount of shape information loss. For instance, the front and rear wheels were blurred—the shape of the font part was not complete. If 12.5% points were dropped, and 128 points remained, only the basic shape and outline were left. Figure 6b was the corresponding recognition performance tested on the different point numbers. It was evident, by that blue curve that represented our proposed DGCB-Net, that it provided better robustness than other popular algorithms. The downward trend of the blue curve was much slower than others, where the yellow and red curves have steeper decline degrees. Especially when samples only own 128 points, our recognition rate was still more than 90.5%. However, that of the DGCNN and PointNet++ were less than 90%.

Thus, the robustness of our DGCB-Net showed better performance when object samples existed a certain degree of point missing.

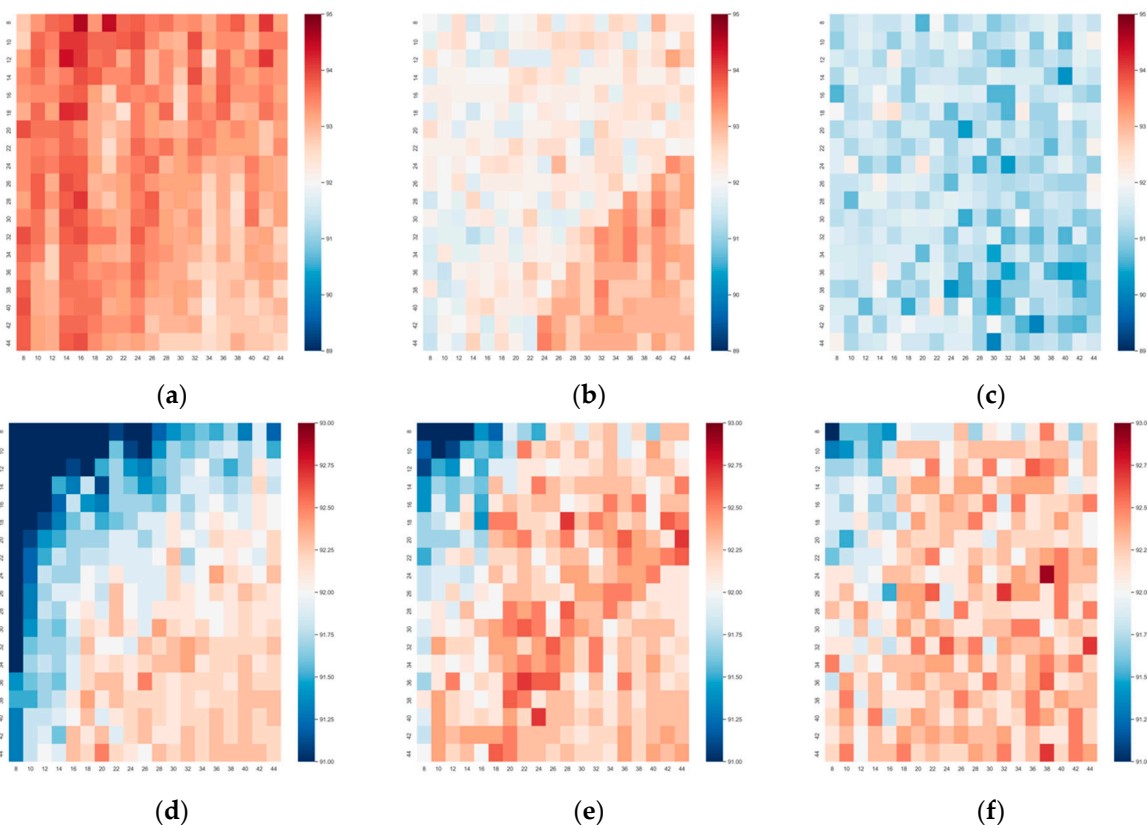

**Figure 5.** Performance of ModelNet10/40 on different broad model size. The horizontal axis is the node number that belongs to each feature layer. The vertical axis is the number of feature layers. Variable N_E means the number of enhancement nodes. MN10 and MN40 represent the dataset ModelNet10 and Modelnet40 respectively. (**a**) N_E = 100 (MN10), (**b**) N_E = 1000 (MN10), (**c**) N_E = 2000 (MN10), (**d**) N_E = 100 (MN40), (**e**) N_E = 1000 (MN40), (**f**) N_E = 2000 (MN40).

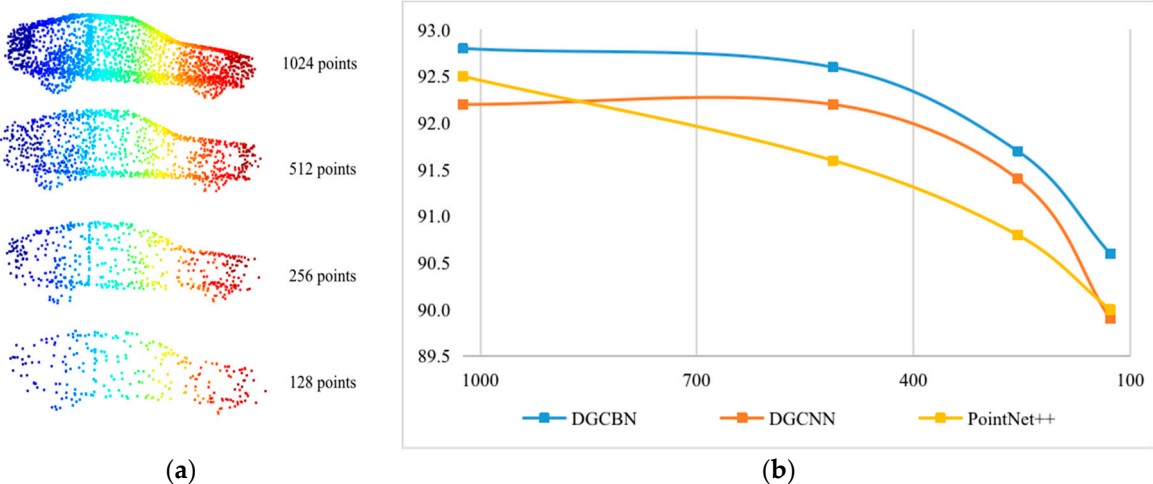

**Figure 6.** Robustness estimation on point cloud with random point dropout based on ModelNet40, (**a**) random point dropout, (**b**) accuracy curve under different point density.

### 4.2. Outdoor Object Datasets

Because the sample count of the Sydney urban object dataset was small, we collected a series of real outdoor LiDAR objects by using our experiment platform. Figure 7 shows the collected 6 kinds of common outdoor objects LSOOD collected by our UGV, which contained cars (a–e), pedestrians (f–i), bushes (j–l), trees (m–p), trunks(q–u), and walls (v–x). The numeric statistic of each kind of object for both training and testing was written in Table 4. There were 1473 samples total, 1010 samples used for model training, and 463 samples for testing. Through using our proposed DGCB-Net, the recognition accuracy reached was around 97.15%.

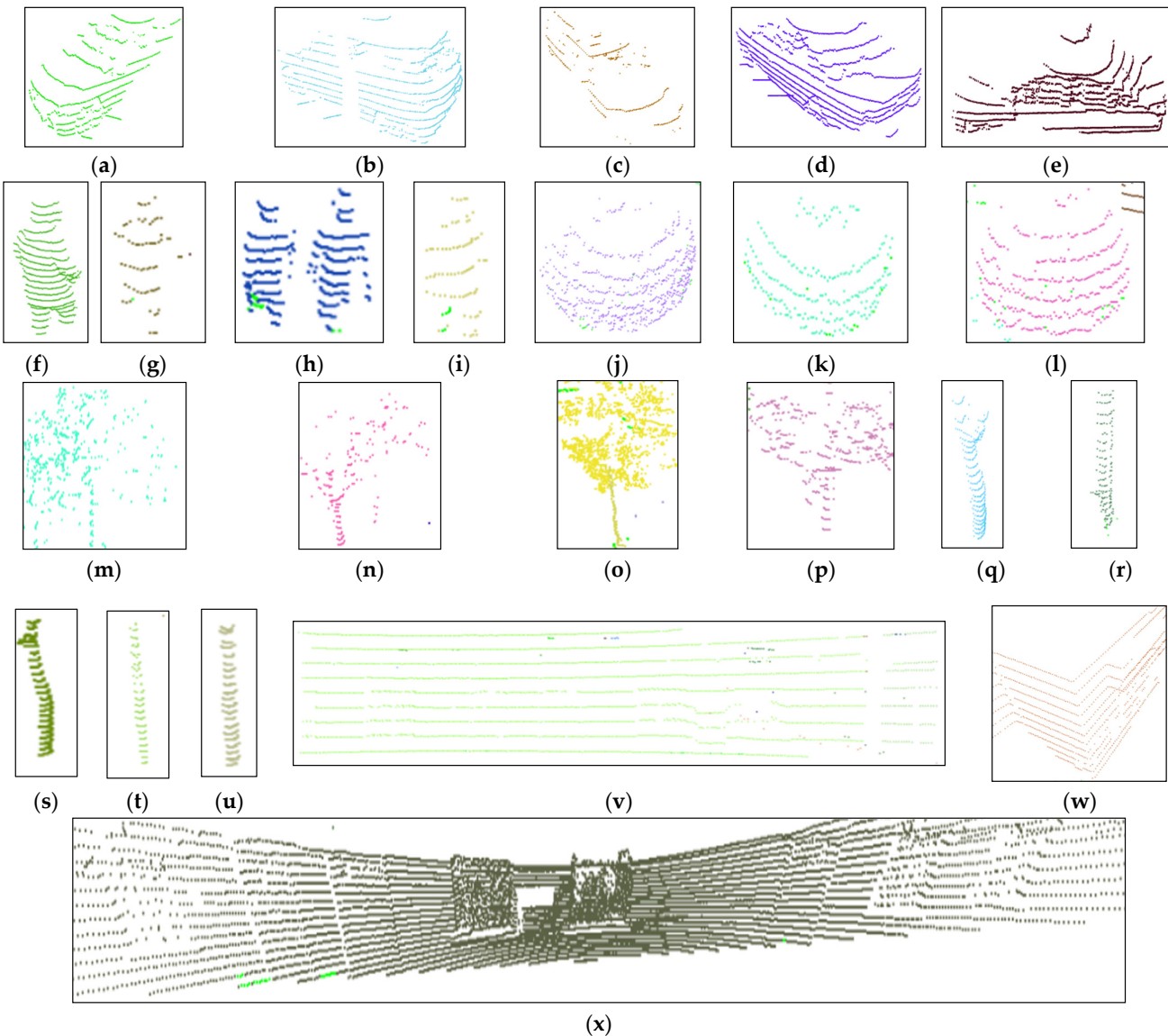

**Figure 7.** Six kinds of object categories are sensed by LiDAR. (**a**–**e**) Car object. (**f**–**i**) Pedestrian object. (**j**–**l**) Bush object. (**m**–**p**) Tree object. (**q**–**u**) Pole-like object. (**v**–**x**) Wall object.

**Table 4.** Outdoor objects collected by our UGV platform.

| Object Type | Pedestrian | Bush | Tree | Trunk | Building | Car | Total |
|---|---|---|---|---|---|---|---|
| **Train** | 100 | 100 | 300 | 150 | 300 | 60 | 1010 |
| **Test** | 52 | 100 | 115 | 86 | 82 | 28 | 463 |
| **Number** | 152 | 200 | 415 | 236 | 382 | 88 | 1473 |

Table 5 gives the detailed precision, recall, and F1 score of each class by the proposed algorithm and the other 3 popular object recognition algorithms. Our algorithm displayed the better performance of F1 scores on bush, tree, building, and car objects. The average weighted precision, recall, and F1 scores reached 0.98.

**Table 5.** Outdoor objects collected by our UGV platform.

| No. | | DGCBN | | | DGCNN | | | PointNet | | | PointNet++ | | |
|---|---|---|---|---|---|---|---|---|---|---|---|---|---|
| | | P | R | F1 | P | R | F1 | P | R | F1 | P | R | F1 |
| 0 | Pedestrian | 0.93 | 0.96 | 0.95 | 0.93 | 1.00 | 0.96 | 0.43 | 0.92 | 0.59 | 0.96 | 0.87 | 0.92 |
| 1 | Bush | 0.98 | 0.96 | 0.97 | 1.00 | 0.95 | 0.97 | 0.98 | 0.76 | 0.86 | 0.92 | 0.98 | 0.95 |
| 2 | Tree | 1.00 | 1.00 | 1.00 | 1.00 | 0.97 | 0.99 | 0.99 | 0.99 | 0.99 | 0.98 | 1.00 | 0.99 |
| 3 | Trunk | 0.95 | 1.00 | 0.98 | 0.97 | 0.99 | 0.98 | 0.99 | 0.99 | 0.99 | 0.92 | 1.00 | 0.96 |
| 4 | Building | 1.00 | 0.98 | 0.99 | 1.00 | 0.93 | 0.96 | 0.99 | 0.99 | 0.99 | 1.00 | 0.97 | 0.99 |
| 5 | Car | 0.99 | 0.96 | 0.98 | 0.84 | 0.97 | 0.90 | 0.98 | 0.98 | 0.98 | 0.99 | 0.92 | 0.95 |
| | avg | 0.98 | 0.98 | 0.98 | 0.97 | 0.96 | 0.96 | 0.97 | 0.95 | 0.95 | 0.97 | 0.97 | 0.97 |

In order to illustrate the robustness of our proposed DGCB-Net on the scanned data, we used a random dropout method to downsample the collected 3D object point cloud. Figure 8a shows the visualization results of the pedestrian, tree, building, and car objects in total, 256, 128, 64, and 32 points, respectively. When the point cloud has 256 points, the object can retain almost all the detailed information. For example, the image in the 1st row and 2nd column can still clearly see the outline of the head and legs. When 128 sampling points were reserved, we can distinguish their categories based on the main parts of object point clouds in the 3rd column. However, if there were only 64 points, pedestrian and tree objects can be vaguely recognized by humans, but the point cloud structure of the car was somewhat illegible. The last column demonstrates the object with only 32 points, the point clouds were difficult to recognize with the naked eye. The recognition performance of the DGCNN model that shows in Figure 8b by the blue curve expresses the obvious fact that fewer number of point clouds own fewer object features. Thus, the slope of the blue curve was very steep. However, the slope of the orange curve was relatively flat even the recognition accuracy was higher than 95% when only 32 points were provided. This way, the experiment can strongly prove that our proposed DGCB-Net algorithm with the broad structure was more robust than the deep learning model DGCNN.

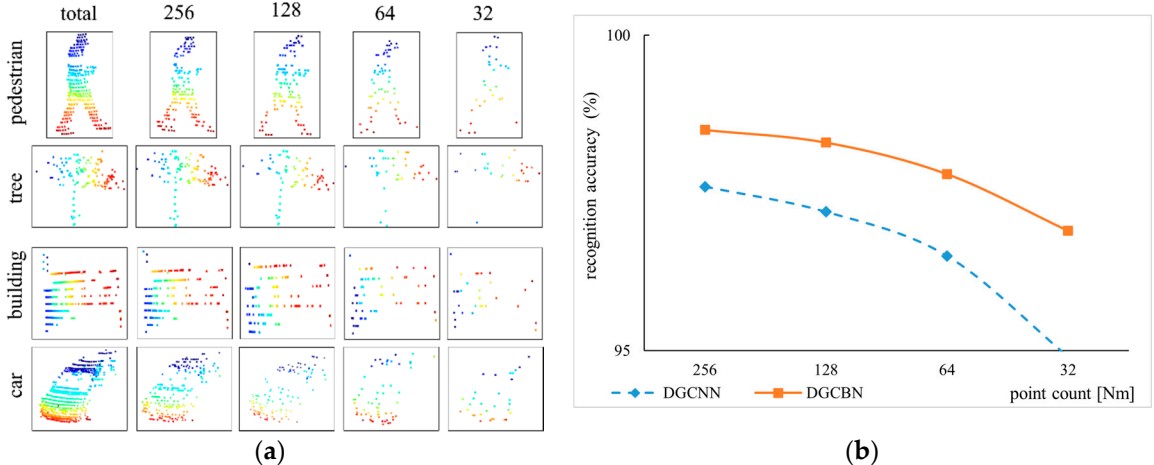

**Figure 8.** Robustness estimation on point cloud with random point dropout based on outdoor object samples. (**a**) Random point dropout, (**b**) Accuracy curve under different point density.

## 5. Discussion

**Dataset Types** This paper estimates DGCB-Net on two different kinds of point clouds. One is the public dataset ModelNet10/40 that consists of evenly distributed point clouds in the whole object space. The other dataset is the outdoor object point clouds scanned by our LiDAR sensor from a certain view that showing sheet structure with occlusion and uneven density. Our proposed DGCB-Net shows better classification performance than other popular object recognition algorithms on both two different kinds of point clouds.

**Performance Improvement** Table 2 clearly shows that our proposed DGCB-Net algorithm has a significant improvement on the public datasets Modelnet10/40. The recognition accuracies of DGCB-Net reach 92.9 and 94.6 on ModelNet10/40, respectively, owning at least 0.5 than other algorithms. Besides, we compared the accuracy of each category in ModelNet40 with the DGCNN algorithm in Table 3. Among them, our DGCB-Net algorithm achieved better efficiency on 17 categories, and 8 categories remained at the same performance as the DGCNN algorithm.

**Broad Network Parameter** We set different counts of nodes and layers in the broad network to explore the relationship between the network size and classification performance. For solving the 10 classification problem on the ModelNet10 dataset, the small parameter size of the broad network reached a better classification performance when comparing Figure 5a–c. However, while solving the 40 classification problem on the ModelNet40 dataset, the big parameter size in the broad network obtained better results, as shown in Figure 5d–f.

**Robustness Discussion** To estimate the robustness of our DGCB-Net, we downsample the point clouds of modelnet40 in Figure 6a and the outdoor dataset collected by ourselves in Figure 8a. When the point cloud is sparse, for example, only 128 points are retained in the modelnet40, the recognition accuracy remains above 90%, as shown in Figure 6b. In the outdoor dataset we collected, although only 32 LiDAR points were retained, the DGCB-Net algorithm can still guarantee higher performance than DGCNN, as shown in Figure 8b.

## 6. Conclusions

This paper proposed a DGCB-Net model to realize object recognition from point clouds by combining convolution and broad learning systems. When estimated on Modelnet40 and our collected outdoor common object datasets, the DGCB-Net model showed a better performance than other state-of-the-art deep learning algorithms. Through graph convolution-like operations and feature concatenation, our broad network obtained descriptive LiDAR point cloud features, which are beneficial for environment perception and consequence scenes reconstruction for UGVs. Besides, the robustness of our algorithm is another significant advantage that the performance is relatively stable even when a certain part of points have been randomly dropped. In the future, more pre-training structures and feature transformation methods will be explored to improve recognition performance. Besides, improving existing deep learning architecture with a flat strengthening way is another interesting research point. Furthermore, other feature extraction models in the deep learning domain will be extended in a flat way to enhance their performance in 3D object recognition and other kinds of point cloud applications.

**Author Contributions:** Methodology, Y.T., W.S.; supervision, L.C.; funding acquisition, Y.S.; investigation, S.W. All authors have read and agreed to the published version of the manuscript.

**Funding:** This research was supported by the MSIT(Ministry of Science, ICT), Korea, under the High-Potential Individuals Global Training Program)(2020-0-01576) supervised by the IITP(Institute for Information & Communications Technology Planning & Evaluation), University of Macau RC MYRG2018-00132-FST, Science and Technology Development Fund, Macao S.A.R (196/2017/A3), University of Macau RC MYRG2018-00132-FST, National Nature Science Foundation of China (No. 61673405 and No. 61503005), the Great Wall Scholar Program (CIT&TCD20190304, CIT&TCD20190305), the National Key R&D Program of China under Grant (2017YFC0821102, 2917YFC0822504), and "Yuyou" Project of North China University of Technology.

**Conflicts of Interest:** The authors declare no conflict of interest.

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
