# Peer review of "DGCB-Net: Dynamic Graph Convolutional Broad Network for 3D Object Recognition in Point Cloud"

_remotesensing, doi:10.3390/rs13010066_

Round 1

Reviewer 1 Report

The work, may include good contributions; but the way it is presented, it cannot be inferred clearly.

> There is a high similarity between the illustrated architecture (Figure 2), and Figure 2 of PointNet [8], even in the details!

> In Figure 2 of PointNet, the term MLP is clearly defined to stand for "Multi-layer Perceptron", but the authors have consistently and wrongly expanded it as "Multiple Layer Perception"!

The above points together arise suspicion!

> The explanation of the method is not clear:

   >> What is the feature extraction part trained on?

   >> What exactly is the "unsupervised" task that is claimed to be used in the introduction?

   >> What is Y (referred to as ground truth).

   >> Most of the section 3 is wasted on explanation of basic ideas of neural networks and fully connected layers -- by the way it is not Full connection layers ! You could instead use the space to explain what you mean by a "broad" network, which seems to be a key component of your work.

> There have been other voxel-based networks which have yielded better results on ModelNet and Sydney. E.g. ORION, others.

Bottom line: I cannot see the contribution. Hopefully this is due to mis-presentation. I recommend the authors to emphasize on their contribution, and explain different steps at a higher level: feature extraction (input-output), aggregation+unsupervised step, etc. You also better skip explanation of the basics of neural networks, such as how a fully connected layer works or how a regularization term takes effect, so the potential reader can have a better understanding of the flow of your work.

Reviewer 2 Report

This paper proposes a neural network approach for identifying 3D shapes out of a 3D point cloud called Dynamic Graph Convolutional Broad Network (DGCB-Net).
The paper shows two main contributions associated with: a) network architecture aiming at improving the object recognition features; b) experimental tests for comparing the proposed method with other state-of-the-art methods.
The paper states clearly from the introduction what the purpose of the work is and its scientific contribution. The experimental results provide a measurable improvement to pre-existing methods.
It is well organized in the theoretical part while it could be improved with minor revisions in discussing the experimental results.
The authors correctly compare their results with those attainable with the ModelNet10/40 and Sydney Urban Object Datasets, but the comparison with newly acquired 3D scans could have been more exhaustive.
In particular, the robustness of the algorithm discussed in lines 443-460 shows the algorithm's behavior in function of different levels of undersampling of the target, which is interesting. But, given the laser scanners' properties, it would also be interesting to check the network's behavior against different levels of 3D uncertainty while maintaining s fixed sampling. This could help compare this performance against different 3D devices, whose random error can change significantly.

I suggest the authors add this kind of test for providing a broader idea of the robustness of the proposed method.

Reviewer 3 Report

In this paper, the authors proposed DGCB-Net for 3D object recognition in point cloud. There are several issues in the current version. First, the novelty is not new. The authors simply combined different existing components into one framework. For example, there is nothing new in Section 3.2. Second, the performance of the proposed framework is not really outstanding. For example, as can be seen in Table 4, DGCB-Net outperforms ECC by 0.01, a very small margin.

Reviewer 4 Report

Great paper, congrats!

Some remarks:

line 32-38: considering the readers' interest fields of MDPI remote sensing I'd extend the first paragraph of the Introduction section with 1-2 sentences on different types of point clouds (ALS, MLS, TLS) to introduce multiple application fields where the proposed methods would effectively improve object recognition.

46: resample the initial point cloud?

Section 2 (Releated works): why past tense is used here?

105-106: consider mentioning the noise as an effect of reflectivity and ranging accuracy

114-116: consider rephrasing the sentence

204: subsections 3.1 and 3.2?

217: is generated?

Figure numbering is wrong, there are two Fig. 3 and three Fig. 4. The cross-references to Figures are also to be corrected in the text!

347: raw point coordinates?

Table 2: the differences between performances in Modelnet10 and Modelnet40 are much higher in the case of image/voxel-based methods compared to graph-based ones. If you have an idea why is that, consider explaining it in the text.

Table: kindly format the heading rows of the tables

505: what is 4wd? 4-wheel-drive car? If yes, why is it relevant in such an investigation? SUV would be more interesting since SUVs have different shapes compared to regular passenger cars...
